# Experimental Study on the Effect of Date Palm Powder on the Thermal and Physico-Mechanical Properties of Gypsum Mortars

Mohamed Aymen Kethiri [1,2,*], Noureddine Belghar [1], Mourad Chikhi [3], Yousra Boutera [4], Charafeddine Beldjani [1] and Cristina Tedeschi [2,*]

1   Laboratoire de Génie Energétique et Matériaux, LGEM, University Mohamed Khider of Biskra,
    Biskra 07000, Algeria; n.belghar@univ-biskra.dz (N.B.); charafeddine.beldjani@univ-biskra.dz (C.B.)
2   Dipartimento di Ingegneria Civile e Ambientale, Politecnico di Milano, 20133 Milan, Italy
3   Unité de Développement des Equipements Solaires, UDES/EPST CDER, Bou-Ismail 42415, Algeria;
    chikhi.mrd@gmail.dz
4   Laboratoire de Génie Mécanique, LGM, University Mohamed Khider of Biskra, Biskra 07000, Algeria;
    yousra.boutera@univ-biskra.dz
*   Correspondence: aymen.kethiri@univ-biskra.dz (M.A.K.); cristina.tedeschi@polimi.it (C.T.)

**Abstract:** Date palm leaves have been diachronically applied in building materials in the Middle East and North Africa (MENA) region, so as to enhance specific properties, such as volume stability and strength. This research work concerns an experimental study on the impact of date palm leaflet powder (DPLP) on the thermal and physico-mechanical properties of gypsum mortars. A series of mortar compositions were prepared with different proportions of DPLP (0, 1, 3, and 5% $w/w$ of binder) and variant particle size (0.5, 1, and 1.5 mm). The results showed that the mortars containing DPLP exhibited significant changes in their properties due to variations in DPLP concentration and particle size. Increased DPLP led to lower density, higher porosity, and water absorption rate, whereas mechanical strength and thermal conductivity were decreased according to the DPLP proportion and size. This research provides valuable insights into the use of sustainable and renewable building materials, highlighting the benefits of exploiting agricultural waste in the constructional sector. The findings lay the groundwork for future research and innovation in environmentally friendly construction technologies.

**Keywords:** date palm; gypsum; mortar; thermal; physico-mechanical properties

## 1. Introduction

Gypsum is a raw material that has been utilized for millennia in construction dating back to the ancient Egyptian civilization [1] and continues to be applied in the MENA (middle East and North Africa) region [2]. It has been utilized in a variety of applications, including plastering, rendering, bedding mortars, stuccowork, and various exceptional decorative techniques for artworks. Gypsum plaster boards are widely used for interior walls and ceilings due to their straightforward manufacturing process, environmental benefits, esthetic appeal, affordability, and excellent properties [3,4].

On the other hand, the reinforcement of building materials with natural fibers dates to prehistoric times, with mud bricks being reinforced with straw or horsehair [5]. To date, numerous studies have aimed to advance composite materials with natural fibers, while research on sustainable building materials have been pursued [6]. The combination of gypsum and date palm leaves represents a promising material due to the leaves' high availability in the MENA region and low environmental impact, making them an appealing choice for ecofriendly building materials [1,3–12].

Recent research has revealed that the number of date palms has exceeded 120 million in the MENA region, with each tree living for more than a century and producing fruits and waste during annual harvests [13,14]. The *Phoenix dactylifera* L. family is home to the

date palm, which is a significant fruit tree in Algeria with over 20 million palms [15,16]. Algeria has the most substantial number of trees in North Africa. Research has shown that the palm waste production exceeds 800,000 tons per year in Algeria alone [17].

Several works have studied the thermophysical, chemical, and dielectric properties of date palm waste and have shown the potential for various applications [12,18] such as its use as a fertilizer in agriculture [12], and also in the automotive [19,20] and construction sectors [5,6,9,21,22]. Awad et al. [13] discussed the potential of date palm fibers to be used as bio-composites for reinforcement in construction materials based on cement, clay, asphalt, and gypsum.

Many researchers have shown that the use of date palm leaves in building materials can have both positive and negative effects on their properties. Boumhaout et al. [22] studied the thermomechanical properties of lime-based mortars reinforced with a date palm fiber mesh. Chennouf et al. [23] evaluated the ability of date palm concrete to dampen the indoor relative humidity while considering the influence of temperature. Feng et al. [24] proposed the material-dependent critical agent concentration (the lowest concentration that ensures hydrophobic effectiveness on a given material), showing that date palm leaves can affect or improve the durability of building materials. Kareche et al. [25] found that natural palm fibers improve the porosity of mortars, increasing the moisture vapor permeability and breathing ability of structures. Bamaga et al. [26] concluded that date palm fibers can be used to improve the physical, mechanical, and thermal properties of concrete and mortar. Finally, Darwish et al. [27] developed low-cost date palm midrib components for lightweight and sustainable long-span multipurpose structures for rural communities, demonstrating the potential of date palm leaves for use in modern construction.

Asim et al. [28] found that the addition of 50% date palm fibers to phenolic composites improved their tensile modulus and impact strength but reduced their tensile and flexural strength. Bamaga et al. [29] found that mortars containing 10 mm and 20 mm date palm fibers had lower water absorption than control mortars, but also lower workability, density, and compressive strength. Chikhi et al. [30] show that DPF loading may have a greater effect on the mechanical and thermal properties of the composites than fiber size. Ali et al. [31] found that date palm leaves and wheat straw fibers can be used to produce thermal insulation materials with a thermal conductivity of 0.045–0.065 W/m. Khoudja et al. [32] found that the admixture of date palm waste to raw clay bricks improved their thermal insulation, with thermal conductivity decreasing from 0.677 W/m K to 0.342 W/m K when 10% DPW was added. Rachedi et al. [5] found that the addition of date palm fibers to gypsum reduced thermal conductivity, thermal diffusivity, and efficiency while increasing the specific heat. Alothman et al. [33] found that composites reinforced with date palm fibers from the trunk and stem of the date palm had better thermal resistance. Finally, Benaniba et al. [9] found that adding date palm fibers to a bio-composite material improved its thermal insulation properties and reduced its thermal conductivity. Gounni et al. [34] developed insulation materials from date palm fibers and cardboard waste and found that they were competitive with conventional insulation materials in terms of thermal properties and economic feasibility. The effects of the size and concentration of date palm leaflet powder on the thermomechanical and hygienic characteristics of a building material such as gypsum have been studied in some studies.

In this work, an experimental study was carried out to assess the impact of the addition of date palm leaflet powder (DPLP) on the thermal and physico-mechanical properties of gypsum mortars for plastering. A series of mortar compositions was prepared with different proportions of DPLP (0, 1, 3, and 5% $w/w$ of binder) and variant particle size (0.5, 1, and 1.5 mm). The aim of this study was to exploit and valorize date palm waste as a renewable, biodegradable material available in the MENA region. Furthermore, it highlights an innovative approach on the application of additives derived from recycled plant waste for the production of lightweight composite materials. By developing lighter construction materials with improved properties, the environmental burden of organic waste will be reduced, as well as the demand of raw materials for construction purposes.

## 2. Materials and Methods

### 2.1. Raw Materials

#### 2.1.1. Gypsum

The gypsum used in this study was obtained from the GCI company in Biskra city in Algeria. According to the company's recommendations, this type of gypsum is preferably used with a water/gypsum ratio (w/g) which can vary from 0.6 to 0.8. The chemical composition of this gypsum is summarized in Table 1.

**Table 1.** Chemical composition of gypsum.

| Constituents | $SO_2$ | CaO | $SiO_2$ | MgO | $AL_2O_3$ | $Na_2O$ | $Fe_2O_3$ |
|---|---|---|---|---|---|---|---|
| Percentage % | 45.63 | 32.12 | 0.58 | 0.41 | 0.12 | 0.09 | 0.08 |

#### 2.1.2. Date Palm Leaflet Powder (DPLP)

The date palm component investigated in this study was obtained from leaves, particularly leaflets (Figure 1b), harvested in the geographic regions of Tolga and Biskra in Algeria. To utilize date palm waste, research focused on the Deglat Noor variety, which has the greatest diversity in this area. The main chemical constituents in date palm leaf powder are cellulose, hemicellulose, and lignin. The powder has low thermal conductivity, as mentioned in references [18,35], which allows it to have the potential to be a reinforcement for composite materials [36]. Sampling was conducted in three distinct phases.

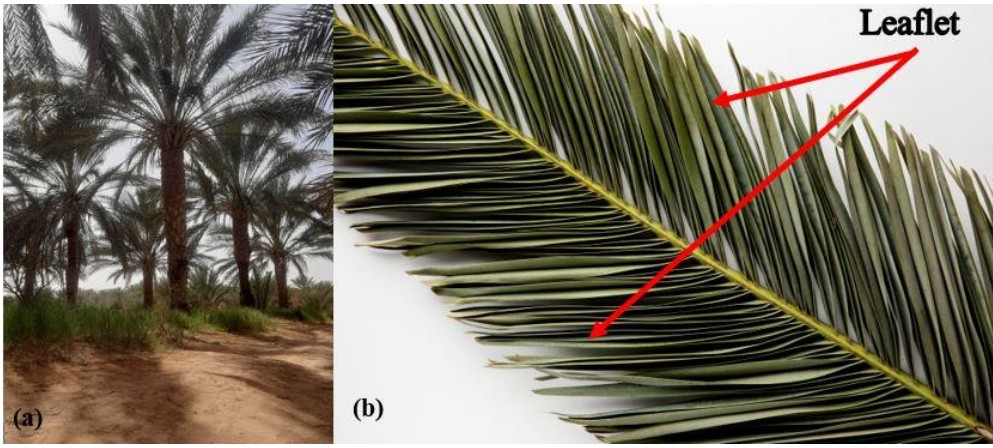

**Figure 1.** (**a**) Date palm tree; (**b**) date palm leaves.

The raw material used in this study was processed according to the procedure used by Kethiri et al. [36]. Date palm leaflets were firstly immersed in a tank of water in order to be cleaned and were afterwards subjected to drying at a temperature of 40 °C 24 h. Then, the dried leaflets were ground using a mechanical grinder up to the selected granulometry.

#### Granulometry

A series of tests were conducted to determine the particle size distribution of the leaflets, while the results were an average of fourth measurements. The particle size distribution was determined by the sieving method according to the French standard 18-560 [37], as shown in Figure 2. The particle size distribution ranged from 0.125 mm and 1.5 mm, whereas the proportion of DPLP particles between 0.125 and 0.25 mm was less than 40%. On the other hand, the proportion of particles between 0.5 and 1.5 mm was more than 85%. The diameters of 0.5, 1, and 1.5 mm represented the dominant part of the date palm powder.

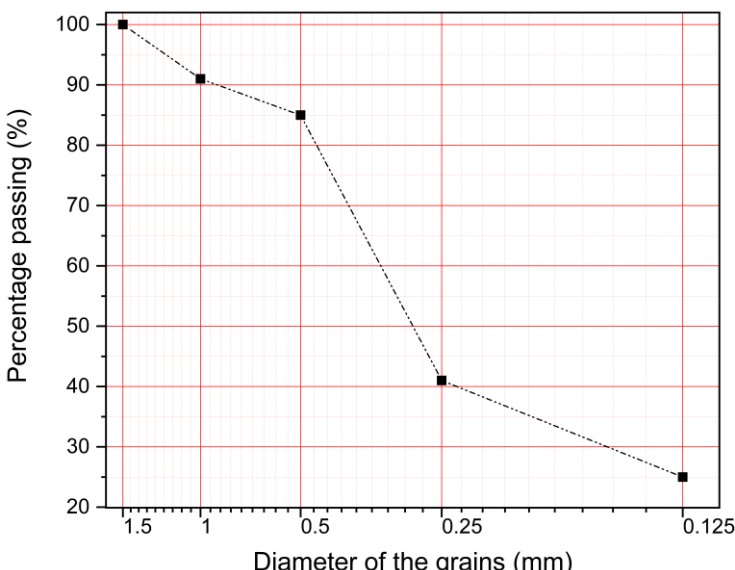

**Figure 2.** DPLP size distribution.

*2.2. Experimental Methodology*

2.2.1. Preparation of Specimens

The dimensions of the powder diameters and the mass fraction of the powder in the mortar varied [38,39]. Dimensions of 0.5, 1, and 1.5 mm were chosen based on the Granulometry test shown in Figure 2, as this interval represented the 90% of the crushed DPLP. The DPLP proportion was 0, 1, 3, and 5 (wt.% of gypsum) for each particle size fraction. Table 2 summarizes the designation used for each composition. During the mortar preparation, several parameters that could influence the results were taken into consideration. The water/gypsum ratio was set at 0.6, according to literature [3–5,30,40].

**Table 2.** Mortar compositions.

| Concentration (wt. %) [Y] | Dimension (mm) [X] | Names |
|---|---|---|
| 0 | / | Reference |
| 1 | 0.5 | G-DPLP-0.5–1 |
| | 1 | G-DPLP-1–1 |
| | 1.5 | G-DPLP-1.5–1 |
| 3 | 0.5 | G-DPLP-0.5–3 |
| | 1 | G-DPLP-1–3 |
| | 1.5 | G-DPLP-1.5–3 |
| 5 | 0.5 | G-DPLP-0.5–5 |
| | 1 | G-DPLP-1–5 |
| | 1.5 | G-DPLP-1.5–5 |

G-DPLP-X–Y, G: gypsum; [X]: dimension (0.5, 1, 1.5 mm); [Y]: concentration (0, 1, 3, 5 wt%).

The preparation of the mortars followed ASTM C1396/C1396M-17 [41]. The mixer used was a TC-MX 1400-2 E (Einhell Company, Nürtingen, Germany) electric mortar mixer. Firstly, the gypsum and DPLP were mixed manually in a container and then the water was added for a total mixing period of 5 min (mixing of 3 min, allowed to rest for 30 s, and then mixed again for 1–2 min). Afterwards it was poured into the molds (4 × 4 × 16 cm) and compaction was made by using a vibrating table for 20 to 30 s. Specimens were stored, as shown in Figure 3, under normal climatic conditions: T = 20 ± 2 °C, and RH = 65 ± 5% for 7, 14 and 28 days.

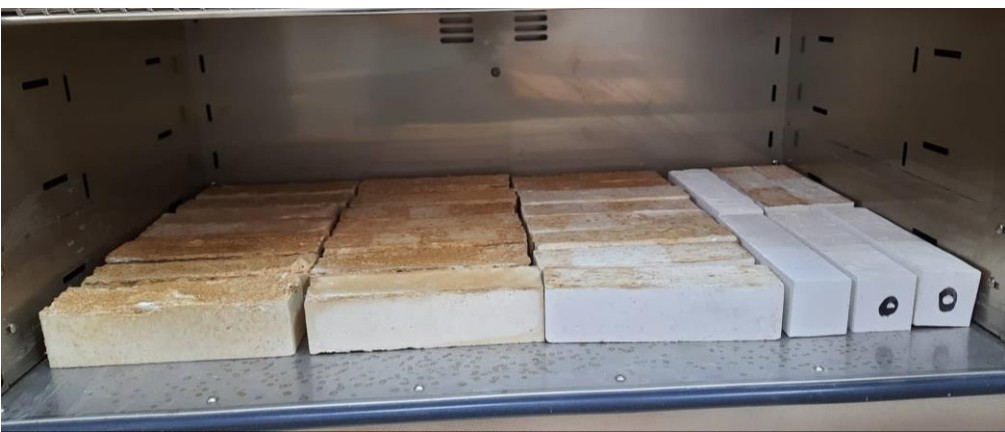

**Figure 3.** Specimens photos.

2.2.2. Testing procedure

Bulk Density, Porosity, and Water Absorption

The density test was carried out on all the specimens, with each one being tested three times. The average was then taken in each case during the calculation.

The apparent density can be determined by using the following relation, according to ASTM C29/C29M [42]:

$$\rho_{\text{app}} = m/V \tag{1}$$

The absolute density: $\rho_{abs}$ is the density without taking into account the voids that may exist in or between the grains and was measured according to ASTM C127/88 [43]. The absolute density was given by Equation (2):

$$\rho_{abs} = \frac{m}{V_2 - V_1} \tag{2}$$

Porosity was calculated by using Equations (1) and (2), as follows:

$$\varnothing = 1 - \frac{\rho_{app}}{\rho_{abs}} \tag{3}$$

Water absorption was carried out on three specimens from each composition (the average was then taken), according to ASTM C642-13 [44], at the age of 7, 14, and 28 days.

Mechanical Characterization

Mechanical strength was tested following ASTM standards. For flexural strength testing (three-point) D790 [45] was applied and for compression C109/C109M-20 [46]. To this direction, an INSTRON 5969 traction machine (Instron Company, Norwood, Massachusetts, United States.) with a capacity of 50 KN and a speed charge of 1 mm/min was employed. Three specimens of each composition were tested at the age of 7, 14, and 28 days.

Thermal Conductivity

Thermal conductivity tests play a crucial role in assessing the heat transfer properties of composite materials, which are used in various applications. For the gypsum/DPLP compositions, ASTM C518-21 [47] was applied. This test method employs a transient heat source and temperature measurement system to determine the thermal conductivity of materials.

## 3. Results and Discussion

### 3.1. Bulk Density and Porosity

In the constructional sector, the utilization of lightweight materials is highly embraced due to their economic advantages [48], due their potential to uphold stability, stiffness,

and durability [7,49]. The density of the examined G-DPLP composites was determined utilizing the five-weighing method. The margin of error in the calculations was below 2%. Figure 4 depicts the variations in absolute and apparent densities, as well as the total porosity of the G-DPLP material about different sizes and mass fractions.

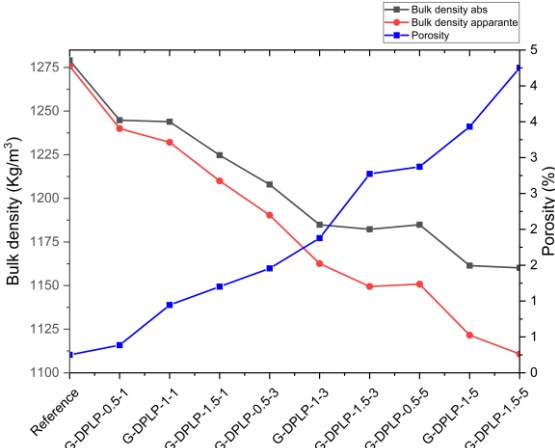

**Figure 4.** The density and porosity of the samples.

Table 3 provides a summary of the densities and total porosity found in both the current study and literature. It was evident that the inclusion of DPLP lightens mortars and leads to a reduction in density up to 13%.

**Table 3.** Density and porosity of samples.

| | **Present Work** | | | **Literature** | |
|---|---|---|---|---|---|
| | **Abs (kg/m$^3$)** | **App (kg/m$^3$)** | **P (%)** | **Bulk Density (kg/m$^3$)** | **References** |
| G-Pure (0%) | 1279.04 | 1275.83 | 0.25 | 1240 | [5] |
| G-DPLP-0.5–1 | 1244.79 | 1240 | 0.38 | | |
| G-DPLP-1–1 | 1243.90 | 1232,14 | 0.94 | 1180 | |
| G-DPLP-1.5–1 | 1224.71 | 1210 | 1.21 | | |
| G-DPLP-0.5–3 | 1207.93 | 1190.35 | 1.45 | | |
| G-DPLP-1–3 | 1184.89 | 1162.64 | 1.87 | 1100 | [4] |
| G-DPLP-1.5–3 | 1182.29 | 1149.5 | 2.77 | | |
| G-DPLP-0.5–5 | 1184.87 | 1150.83 | 2.87 | | |
| G-DPLP-1–5 | 1161.45 | 1121.59 | 3.43 | 1000 | |
| G-DPLP-1.5–5 | 1160.15 | 1110.83 | 4.25 | | |

The lightness of the tested compositions can be ascribed to the alveolar arrangement of DPLP and its porous nature. The inclusion of DPLP, whether in terms of weight or size, results in a rise in the porosity of the mixtures. Gypsum mortars that contains DPLP at various fiber sizes demonstrates higher porosity in comparison to pure gypsum [40,50]. This could be attributed to the formation of voids at the interfacial areas between the DPLP and the gypsum matrix. The obtained outcome concurs with relevant studies in the literature [51,52], according to which the introduction of DPLP into the matrix amplifies the degree of air entrapment. The variability in porosity values can be attributed to the proportion and dimensions of DPLP, as highlighted by [53,54]. Furthermore, the porosity was influenced by the proportion and particle size of DPLP, as well as its distribution within the gypsum matrix. Chikhi [3] elaborated upon a similar trend.

### 3.2. Water Absorption

The water resistance of composites plays a crucial role, particularly when natural fibers are utilized. Various factors influence how composite materials absorb water, including temperature, additives volume fraction and type, area of exposed surfaces, interfacial

bonding, diffusivity, reaction between water and matrix, surface protection, voids, and hydrophobic chains of the matrix [55,56]. Based on Figure 5, it was evident that water absorption increased over time, and all the curves exhibited a similar pattern. Additionally, the reference mortar (with no DPLP addition) had a lower adsorption rate compared to all other compositions.

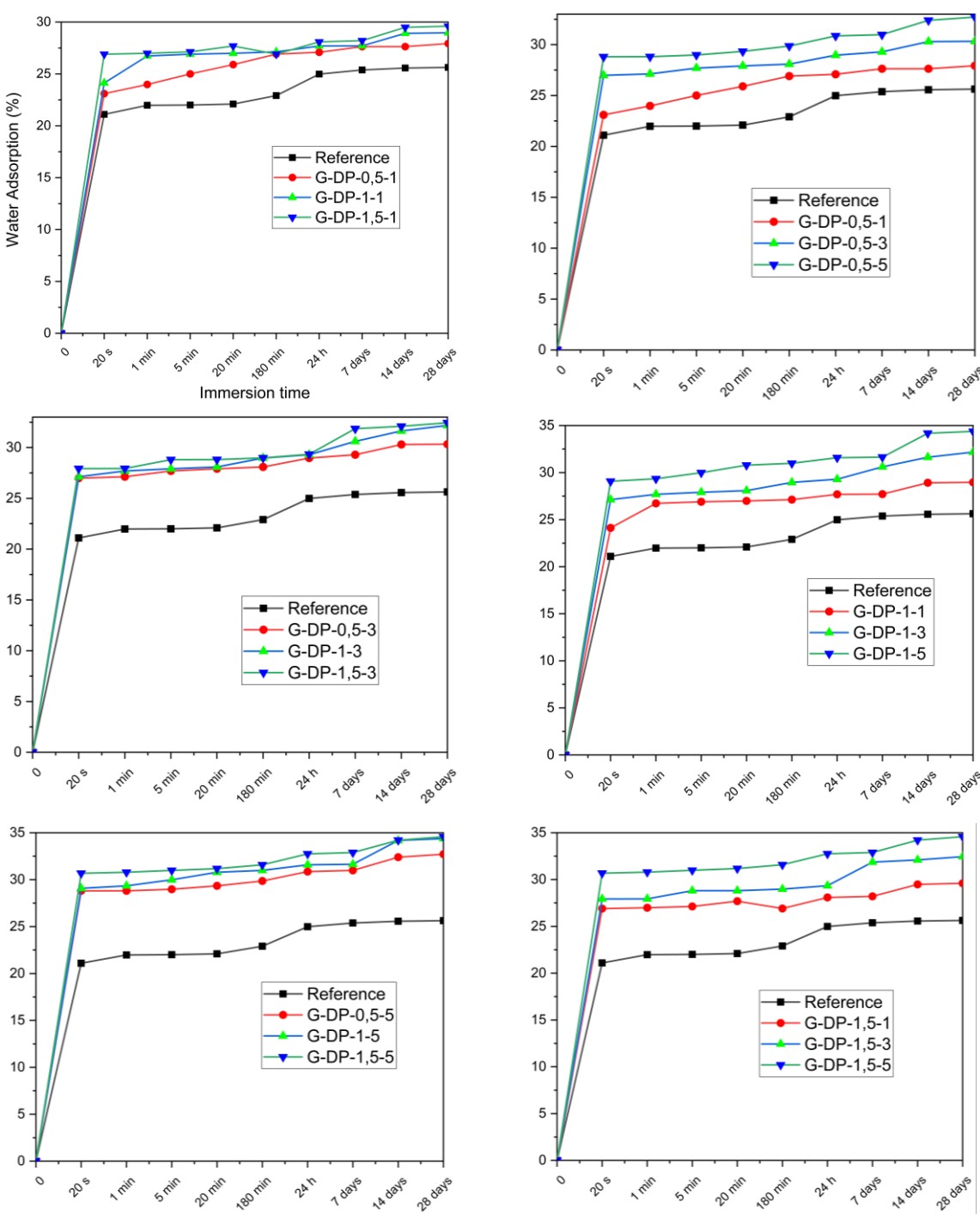

**Figure 5.** Water adsorption in the function of immersion time of composite samples.

A noteworthy observation was that water absorption primarily occurs during the initial stages of immersion, with a substantial quantity being absorbed within the first twenty seconds [57]. The absorption process was largely influenced by the open porosity of the material. The introduction of DPLP led to the formation of numerous pores [58,59].

It was widely acknowledged that lingo-cellulosic materials have a hydrophilic nature due to the presence of cellulose, hemicellulose, lignin, and other factors that contribute to their moisture absorption from the surrounding atmosphere [60–62]. Furthermore, the utilization of DPLP results in higher water absorption. Notably, the concentration of DPLP has a greater impact on water absorption compared to variations in diameter.

Figure 6 shows the evolution of the water content of gypsum-based composites filled with three different mass fractions and three sizes of DPLP on days 7, 14, and 28. It appears that the addition of DPLP with these different sizes in gypsum induces an increase in water content (7.51 to 8.94%). For the same composition, there was a slight increase in water content between day 7 and day 14, and between day 14 and day 28, with an increase of between 0.19 and 2.5 and between 0.06 and 0.5, respectively. This increase was greater when DPLP was added. These results indicate that the absorption rate decreases with time, confirming the previous results showing that absorption takes place in the early stages. These results indicate that incorporating a small amount of DPF into gypsum enhances water absorption [63].

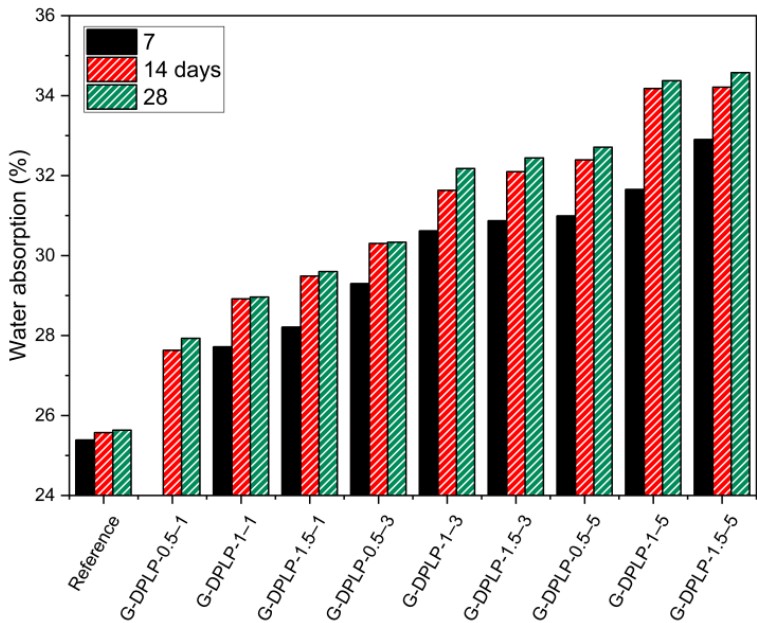

**Figure 6.** Water adsorption test after 7, 14, and 28 days of immersion of composite samples.

Figure 7 represents the water absorption of G-DPLP composites after 28 days. From the results, it was observed that the porosity of the mortar led to a significant increase in the values. This was due to the addition of DPLP and its hygroscopic physiognomy [30]. Chikhi et al. [30] stated that the water content of composites depends strongly on the DPLP water absorption capacity. According to numerous authors, the absorption depends on the size, content, and chemical composition (cellulose) of the natural fiber [9,64,65].

*3.3. Mechanical Proprieties*

The addition of DPLP into the compositions significantly impacts its mechanical behavior, particularly in terms of flexural and compressive.

3.3.1. Flexural Strength

Figure 8 depicts the flexural strength for all specimens at the age of 7, 14, and 28 days. The obtained outcomes reveal that flexural strength diminishes as the dimensions and concentration of DPLP increase [9,66]. A reduction of 25% in flexural strength was observed upon the inclusion of 1% DPLP, followed by a declining fraction of decrease as the concentration or dimensions increase, within the range of 5.3–1.7%. Furthermore, a rise in flexural strength can be observed over time, specifically from 7 to 14 to 28 days. The

reference composition showed a strength increase of 13% from 7 to 28 d, which represents the most substantial elevation compared to the other compositions (5–8%).

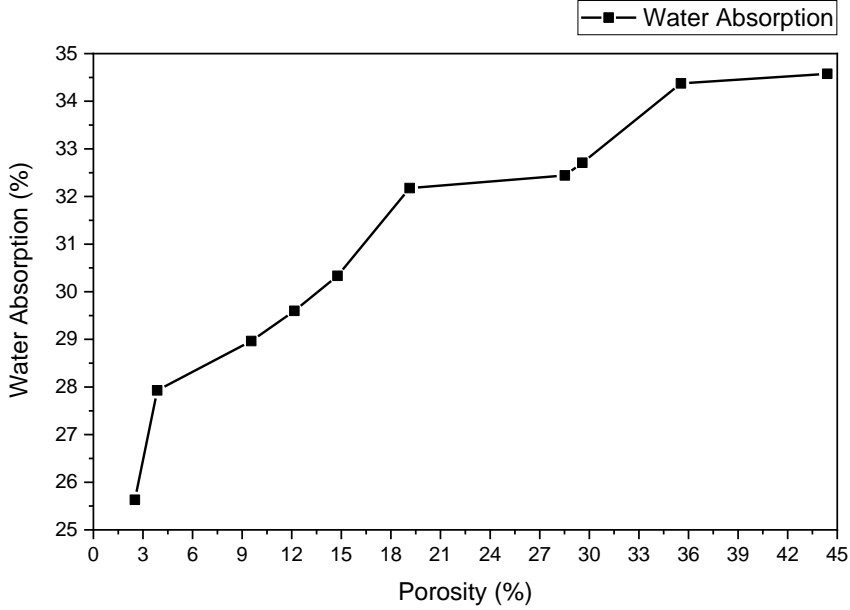

**Figure 7.** Correlation of water absorption with porosity of the compositions.

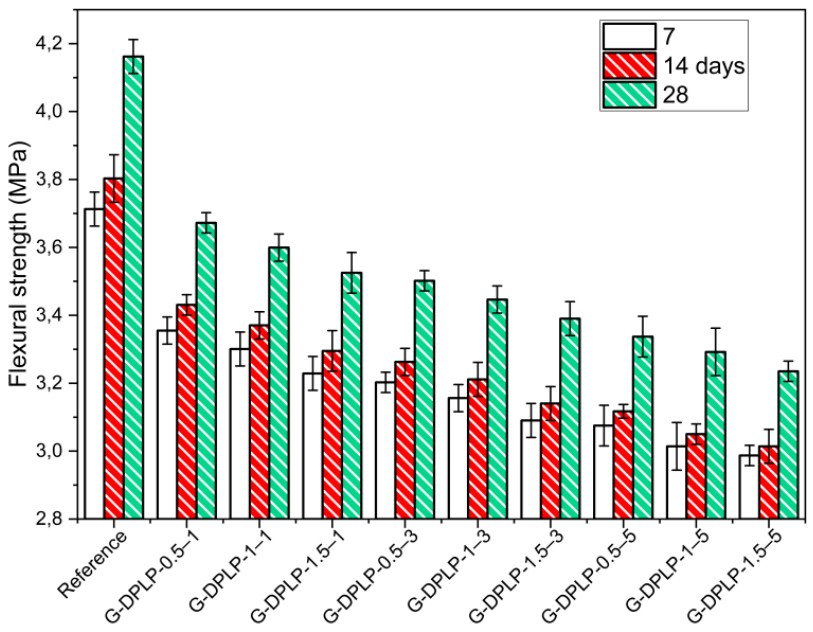

**Figure 8.** Flexural strength results of tested composites after (7, 14, and 28 days).

### 3.3.2. Compressive Strength

Figure 9 shows the compressive strength for all compositions at the age of 7, 14, and 28 days, which was decreased, while the dimension and concentration of DPLP increased [67,68]. A drop of 37% was shown when adding 3% of DPLP and was further decreased (up to 50%) in a concentration of 5% of DPLP. Over time, within a period of 7–14 to 28 days, an increase in compressive strength can also be observed, with an increase of 16% in the case of the reference mortar and an increase of 3–12% in the modified mortars. The latter variation depends on the proportion and particle size of DPLP (decreases with the increase in concentration or dimension).

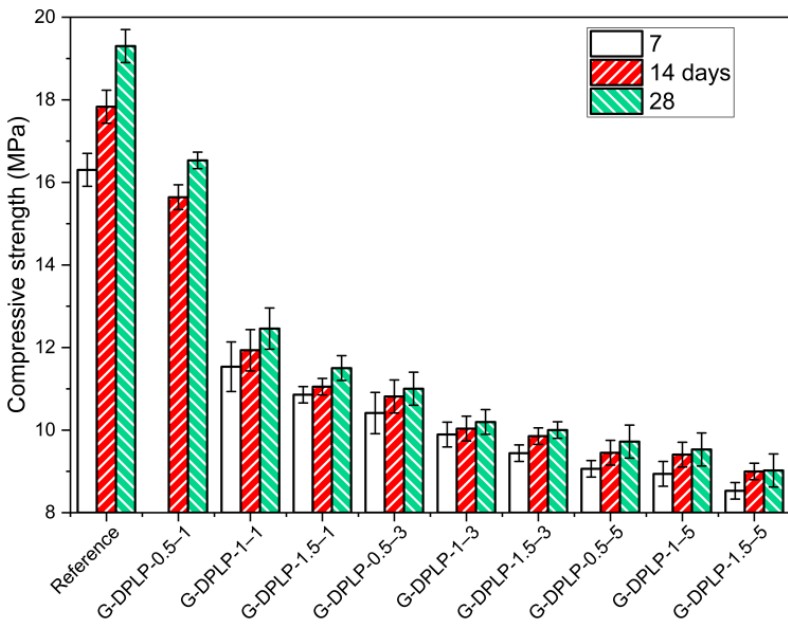

**Figure 9.** Compression strength results of tested composites after (7, 14, and 28 days).

### 3.4. Thermal Conductivity

Thermal conductivity represents a crucial characteristic within the domain of thermal insulation materials. This particular property is predicated upon a multitude of factors, including the morphology, density, and homogeneity of materials [61]. The thermal conductivity's variation, correlated with the porosity of the compositions, is illustrated in Figures 10 and 11. This variation was examined for three different DPLP sizes and concentration. The utilization of gypsum either in isolation or in conjunction with other materials enhances insulation due to its notably low thermal conductivity, which amounts to 0.8 W/m·K in the context of our study [69].

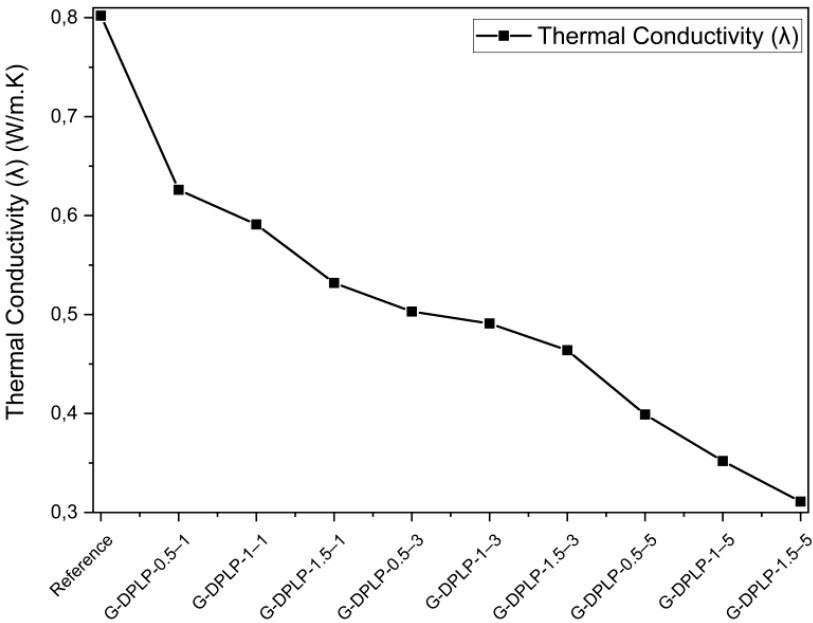

**Figure 10.** Evolution of the thermal conductivity of tested composites.

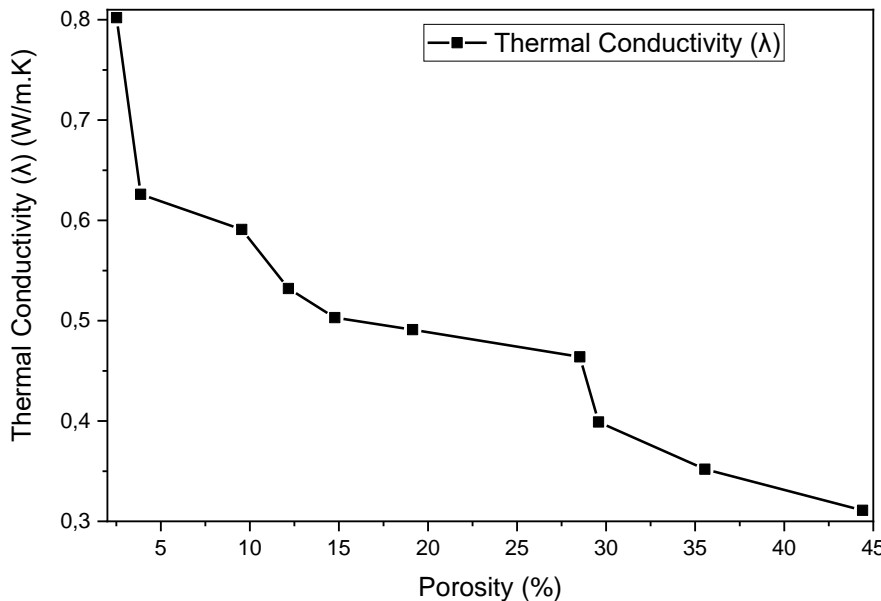

**Figure 11.** Correlation of thermal conductivity with the porosity of the material studied.

Over 28 days, the thermal conductivity diminishes as the dosage of DPLP increases. Moreover, it decreases in tandem with the rise in porosity rate and was inversely proportional to the voids that generated by DPLP presence [70]. The relationship between thermal conductivity and DPLP was closely intertwined. Consequently, as the quantity of DPLP increases, thermal conductivity experiences a decline [62,71,72].

In the present study, when adding 1% of DPLP, thermal conductivity decreased around 22%, compared to the reference mortar, representing the most significant change. With each increase in DPLP concentration or size, leading to a respective porosity rise, thermal conductivity was decreased. The recorded decrease ranges from 2 to 14%. Experimental results demonstrate that the thermal insulation effectiveness of the material was inversely related to its density [50]. These findings highlight the potential for producing insulation materials modified with DPLP.

## 4. Conclusions

This study focused on assessing the effect of DPLP particle size and concentration on the thermal and physico-mechanical properties of gypsum mortars. The addition of DPLP modifies the gypsum matrix, creating voids and increasing porosity. The main conclusions deriving from this study are as follows:

1. An analysis of the size of the granules of DPLP shows that the proportion of particles between 0.5, 1, and 1.5 mm was greater than 85%.
2. As the concentration and size of DPLP increases, the absolute and apparent densities proportionally decrease. The decrease in density was directly related to an increase in porosity. These observations reveal a complex relationship between the amount and size of DPLP, providing important insights into the material's structural characteristics.
3. Water absorption notably increases when DPLP is added, primarily occurring at the beginning of testing. This highlights the significant impact of DPLP on the material's ability to absorb water quickly.
4. The mechanical properties of the modified mortars decreased significantly with increasing DPLP concentration and size. This was due to the voids created by the addition of DPLP, making the material more brittle and porous. Thus, the concentration of DPLP and the level of porosity are crucial factors affecting the structural integrity of the material.
5. The thermal conductivity was decreased by the increase in the DPLP concentration and size. This suggests that thermal conductivity was inversely proportional to the

voids created by the addition of DPLP; indeed, the voids created represent additional thermal resistance, which explains the reduction in thermal conductivity.

According to the experimental results, porosity was the parameter that mostly influenced the physical, mechanical, and thermal properties of the tested mortars. DPLP addition may modify the porosity of the gypsum mortars, according to its proportion and particle size. An increase in porosity imparts a degree of lightness to the material, favorable for specific applications (i.e., renders and plasters). To this extend, DPLP can be used for producing sustainable mortars with elaborated properties, following the diachronic constructional principles of the MENA region.

**Author Contributions:** Conceptualization, investigation, resources, writing—original draft preparation, M.A.K. and N.B.; visualization, supervision, M.C.; supervision, visualization, writing—review and editing, Y.B.; writing—review and editing, C.B.; supervision, visualization, writing—review and editing, C.T. All authors have read and agreed to the published version of the manuscript.

**Funding:** This project work is supported by MESRS of Algeria, and it is funded by the Laboratoire de Génie Energétique et Matériaux, Dipartimento di Ingegneria Civile e Ambientale (DICA), Politecnico di Milan, Solar Equipment Development Unit, and Directorate General of Scientific Research and Technological Development-DGRSDT.

**Institutional Review Board Statement:** Not applicable.

**Informed Consent Statement:** Not applicable.

**Data Availability Statement:** Data are available from the authors.

**Acknowledgments:** Huge thanks are due to the Directorate General of Scientific Research and Technological Development-DGRSDT for its support.

**Conflicts of Interest:** The authors declare no conflicts of interest.

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
