# Peer review of "Experimental Study on the Effect of Date Palm Powder on the Thermal and Physico-Mechanical Properties of Gypsum Mortars"

_sustainability, doi:10.3390/su16073015_

Round 1

Reviewer 1 Report

Comments and Suggestions for Authors

Dear Authors,

The manuscript submitted for review is interesting, but requires a few key corrections. Detailed comments below.

Line 174: At the end of the introduction, you should precisely specify your research goal. The scientific goal should be preceded by a justification for undertaking such research.

Line 175: I admit that I prefer the description of experimental research to have the description "research methodology". The next subsections are subchapters. However, this is a typical experimental article.

Line 176 to 184: This passage should be moved to the end of the "introduction", after the scientific purpose.

Line 198: Write what fraction range the raw material will have after grinding. You can also provide the size of the sieve on which the raw material was sifted.

Provide the name of the dryer, grinder and add a description: (name: manufacturer, city, country). Every scientific equipment used in research should be described in this way. Please review the entire research methodology in this regard.

Line 204: Enter standard deviation/temperature error +/- setpoints.

Line 326: It is not written how many times each experiment was repeated.

Line 327: There are no error whiskers in the charts. In fact, they should be on almost all charts. There is no statistical analysis of the research results in the article, which is why the error whiskers are important.

Line 327: Discussion of test results is too general. Some of the descriptions fit more into the introduction than into the description of research results. Especially the first sentences of subsections. You should review the entire Research Results and Discussion section.

Figure 4b: The results refer to individual materials, so the lines between the points should not be connected.

Line 470: Don't write anything about the purpose of the research here. This needs to be included in the introduction.

The conclusions contain many statements and fewer conclusions. Add one more forward-looking conclusion summarizing your work.

Author Response

Detailed responses to the reviewer 1 comments

Manuscript ID: sustainability-2905412

Title: Experimental study on the porosity effect on thermomechanical and hydric characteristics for building material based on date palm.

We thank reviewer 1 for his valuable remarks and comments. We have provided more details and illustrations in the revised paper version. All comments have been considered. Knowing that the authors made some modifications in the paper to improve the paper.

Please find our responses to reviewer comments here.

Note: the changes made in the text are in red

Reviewer #1:

The manuscript submitted for review is interesting but requires a few key corrections. Detailed comments below.

We thank reviewer 1 for their positive comments and recommendations. Grammatical and scientific errors in the manuscript have been corrected. The article is now carefully edited, and the suggestions recommended by the reviewer are now reflected in the revised manuscript. If there is any other comment, it is available to redo them.

Comments:

  • Line 174: At the end of the introduction, you should precisely specify your research goal. The scientific goal should be preceded by a justification for undertaking such research.

Answer: We thank reviewer #1 for these positive comments. We totally agree with you and your comments help us to correct and to improve the introduction. Below you will find our new expression.

‘The aim of this study was to exploit and valorise date palm waste as a renewable, biodegradable material available in MENA region. Furthermore, it highlights an innovative approach on the application of additives derived from recycled plant waste for the production of lightweight composite materials. By developing lighter construction materials with improved properties, the environmental burden of organic waste will be reduced, as well as the demand of raw materials for construction purposes.’

  • Line 175: I admit that I prefer the description of experimental research to have the description "research methodology". The next subsections are subchapters. However, this is a typical experimental article.

Answer: We thank reviewer #1 for the positive remark. We took your comment into consideration, and it helped us to really enhance and improve the structure. You can see the change in the article.

  • Line 176 to 184: This passage should be moved to the end of the "introduction", after the scientific purpose.

Answer: We thank reviewer #1 for the positive comments. We totally agree with you and your comments help us to correct and to improve the introduction. The passage mentioned was deleted.

  • Line 198: Write what fraction range the raw material will have after grinding. You can also provide the size of the sieve on which the raw material was sifted.

Answer: We thank reviewer #1 for these positive comments. We took your comment into consideration. You can find the information required in the '2.1.2.1. Granulometry' parts. Also, you can observe the all the information asked in the text below.

‘The particle size distribution ranged from 0.125 mm and 1.5 mm, whereas the proportion of DPLP particles between 0.125 and 0.25 mm was less than 40%. On the other hand, the proportion of particles between 0.5 and 1.5 mm was more than 85%. The diameters of 0.5, 1, and 1.5 mm represented the dominant part of the date palm powder.’

  • Provide the name of the dryer, grinder and add a description: (name: manufacturer, city, country). Every scientific equipment used in research should be described in this way. Please review the entire research methodology in this regard.

Answer: We thank reviewer #1 for the positive comments. We have taken your comment into consideration. You can also observe the corrections and information in the revised version.

  • Line 204: Enter standard deviation/temperature error +/- setpoints.

Answer: We thank reviewer #1 for the positive comments. The passage was deleted to improve the quality of paper and to put the most essential for improving the paper.

  • Line 326: It is not written how many times each experiment was repeated.

Answer: We thank reviewer #1 for these positive comments. We have taken your comment into consideration. You can also observe the corrections and information requested in each test.

  • Line 327: There are no error whiskers in the charts. In fact, they should be on almost all charts. There is no statistical analysis of the research results in the article, which is why the error whiskers are important.

Answer: We thank reviewer #1 for these positive comments. We have redesigned all the graphics that require based on your remarks.

  • Line 327: Discussion of test results is too general. Some of the descriptions fit more into the introduction than into the description of research results. Especially the first sentences of subsections. You should review the entire Research Results and Discussion section.

Answer: We thank reviewer #1 for these positive remarks. We have taken your comments into consideration and are revised the discussion.

  • Figure 4b: The results refer to individual materials, so the lines between the points should not be connected.

Answer: We thank reviewer #1 for these positive comments. The results refer to the same materials but with different mass fractions and dimensions.

  • Line 470: Don't write anything about the purpose of the research here. This needs to be included in the introduction.

Answer: We thank reviewer #1 for these positive comments. We have taken your comments into consideration and are revised the discussion.

  • The conclusions contain many statements and fewer conclusions. Add one more forward-looking conclusion summarizing your work.

Answer: We thank reviewer #1 for these positive remarks. We have taken your comments into consideration and are revised the discussion.

We thank reviewer 1 for his valuable remarks and comments.

Sincerely.

Reviewer 2 Report

Comments and Suggestions for Authors

The paper investigates the effects of incorporating date palm leaf powder (DPLP) into gypsum-based composites on properties such as bulk density, porosity, water absorption, mechanical properties, and thermal conductivity. By analyzing different concentrations and sizes of DPLP, the study reveals that DPLP inclusion reduces density, increases porosity, and enhances water absorption while adversely affecting mechanical strength. However, DPLP addition also decreases thermal conductivity, making the material potentially suitable for specific industrial applications. Overall, the research aims to develop more sustainable and durable building materials by leveraging DPLP's properties and provides insights into material science for future advancements in the field.

Unfortunately, the manuscript cannot be accepted in its current form.

1.     What are the main chemical constituents in Date Palm Leaf Powder, and how do they contribute to its properties and potential applications in composite materials?

2. The primary composition is expected to be cellulose; however, it has been reported that various minerals, such as silicon or silicon oxide, may also be present. These materials could interact with gypsum over the long term, forming various mineral phases. Therefore, the authors need to report the chemical composition of the powder being used for the additive/composite.

3.     What is the appearance of the material? The authors should provide photographs of the material. The physical appearance and some properties, such as color, can be relevant in building materials.

4. Materials with lower water absorption are generally considered preferable in construction due to their enhanced durability against aggressive substances and improved material strength. Standard concrete and cement mortar typically exhibit a water absorption rate of around 5%, with values exceeding 10% considered unacceptable. However, high-performance concrete or other specially designed composites may achieve water absorption rates lower than 3%. Authors should clarify the potential applications of this material given its relatively high water absorption percentages.  The authors should discuss the specific potential applications of the material.

5.     How many samples were tested in each of the experiments? The manuscript completely lacks statistical treatment of the measurements and results. The authors should incorporate statistical analysis into the experimental results to ensure reproducibility.

6.     The species' name must be in italics: Phoenix dactylifera L.

 7.       On line 17, page 1, define the first time the acronym DPLP appears.

8.       Table 4 should homogenize the significant numbers used to report each density value. If the measurements were made with the same instruments, they must have the same precision and significant numbers

Author Response

Detailed responses to the reviewers’ comments

Manuscript ID: sustainability-2905412

Title: Experimental study on the porosity effect on thermomechanical and hydric characteristics for building material based on date palm.

We thank reviewer 2 for his valuable remarks and comments. We have provided more details and illustrations in the revised paper version. All comments have been considered. Knowing that the authors made some modifications in the paper to improve the paper.

Note: the changes made in the text are in red

Reviewer #2:

The paper investigates the effects of incorporating date palm leaf powder (DPLP) into gypsum-based composites on properties such as bulk density, porosity, water absorption, mechanical properties, and thermal conductivity. By analyzing different concentrations and sizes of DPLP, the study reveals that DPLP inclusion reduces density, increases porosity, and enhances water absorption while adversely affecting mechanical strength. However, DPLP addition also decreases thermal conductivity, making the material potentially suitable for specific industrial applications. Overall, the research aims to develop more sustainable and durable building materials by leveraging DPLP's properties and provides insights into material science for future advancements in the field. Unfortunately, the manuscript cannot be accepted in its current form.

We thank reviewer 2 for their positive comments and recommendations. Grammatical and scientific errors in the manuscript have been corrected. The article is now carefully edited, and the suggestions recommended by the reviewer are now reflected in the revised manuscript. If there is any other comment, it is available to redo them.

Comments:

  • What are the main chemical constituents in Date Palm Leaf Powder, and how do they contribute to its properties and potential applications in composite materials?

Answer: We thank reviewer #2 for these positive comments. We have taken your comment into consideration, and we add the chemical composition in revised version. you can observe the add part below. You will find the contributions of these chemical compositions in the discussion sections.

‘The main chemical constituents in Date Palm Leaf Powder are cellulose, hemicellulose and lignin. The powder has low thermal conductivity, as mentioned in reference [18],[35], this allows it to have the potential to be a reinforcement for composite materials [36].’

  • The primary composition is expected to be cellulose; however, it has been reported that various minerals, such as silicon or silicon oxide, may also be present. These materials could interact with gypsum over the long term, forming various mineral phases. Therefore, the authors need to report the chemical composition of the powder being used for the additive/composite.

Answer: We thank reviewer #2 for these positive comments.

The chemical composition of the powder used for the additive/composite is mentioned in Table 1 and that of the gypsum is mainly composed of CaSO4 and 0.5H2O [1]. Furthermore, the quantity of CaO and SO2 constitutes the majority of the constituent elements of the powder. For this purpose, the mixture can develop other mineral formulations.

  • What is the appearance of the material? The authors should provide photographs of the material. The physical appearance and some properties, such as color, can be relevant in building materials.

Answer: We thank reviewer #2 for these positive comments. We have taken your comment into consideration. We insert a photo of the material.

  • Materials with lower water absorption are generally considered preferable in construction due to their enhanced durability against aggressive substances and improved material strength. Standard concrete and cement mortar typically exhibit a water absorption rate of around 5%, with values exceeding 10% considered unacceptable. However, high-performance concrete or other specially designed composites may achieve water absorption rates lower than 3%. Authors should clarify the potential applications of this material given its relatively high-water absorption percentages.  The authors should discuss the specific potential applications of the material.

Answer: We thank reviewer #2 for these positive comments.

‘The composite developed in this study is intended for thermal insulation in buildings, generally used as interior cladding. the presence of water increases the thermal conductivity of the composite and its density, reducing the insulation performance of the composite. For this purpose, the composites are dried in order to eliminate the maximum water, because the two materials subject to our study, gypsum and natural fibers, absorb a lot of water in the preparation phase.’

  • How many samples were tested in each of the experiments? The manuscript completely lacks statistical treatment of the measurements and results. The authors should incorporate statistical analysis into the experimental results to ensure reproducibility.

Answer: We thank reviewer #2 for these positive comments. We have taken your comment into consideration. You can also observe the corrections and information requested in each test.

  • The species' name must be in italics: Phoenix dactylifera L.

Answer: We thank reviewer #2 for these positive comments. We have taken your comment into consideration. You can also observe the corrections and information requested in each test.

  • On line 17, page 1, define the first time the acronym DPLP appears.

Answer: We thank reviewer #2 for these positive comments. We have taken your comment into consideration. You can also observe the corrections and information requested in each test.

  • Table 4 should homogenize the significant numbers used to report each density value. If the measurements were made with the same instruments, they must have the same precision and significant numbers.

Answer: We thank reviewer #2 for these positive comments. We have taken your comment into consideration. You can also observe the corrections and information requested in each test.

We thank reviewer 2 for his valuable remarks and comments.

Sincerely,

Round 2

Reviewer 1 Report

Comments and Suggestions for Authors

You have made significant corrections to your article. Compared to the previous version, this article is much better.

My main comment concerns only the lack of error whiskers, on some plots. You should clarify this and, if possible, make corrections. Afterwards, it accepts all corrections made.

Author Response

Thank you very much for your comments and for your acceptance; It’s a pleasure to correspond with you again.

I added the error whiskers to the mechanical property curves per your suggestion. However, I chose not to include them in density, porosity and thermal conductivity because the error was negligible and I thought if I added they would become cluttered and compromise readability. I preferred to maintain clarity and visibility in these plots.

Thank you again for your valuable comments.

Reviewer 2 Report

Comments and Suggestions for Authors

The authors have responded satisfactorily to all my requests and have improved the manuscript. I recommend the publication

Author Response

Thank you very much for your kind words and for your acceptance of the revised manuscript. Your comments and remarks has been invaluable in improving the quality of our work. We greatly appreciate your thorough review and are pleased to hear that you found our responses satisfactory.

Your recommendation for publication is a significant vote of confidence, and we are grateful for your support. We are committed to continuing our efforts to produce high-quality research.

Thank you once again for your time, effort, and positive evaluation.